# Character-Driven Narrative Generation for Scene-Based Video Synthesis

## Abstract

Recent advances in scene-based video generation have enabled systems to synthesize coherent visual narratives from structured prompts. However, a crucial dimension of storytelling—*character-driven dialogue and speech*—remains underexplored. In this paper, we present a modular pipeline that transforms action-level prompts into visually and auditorily grounded narrative dialogue, enriching visual storytelling with natural voice and character expression. Our method takes as input a pair of prompts per scene, where the first defines the setting and the second specifies a character's behavior. While a story generation model such as Text2Story produces the corresponding visual scene, we focus on generating expressive, character-consistent utterances grounded in both the prompts and the scene image. A pretrained vision-language encoder extracts high-level semantic features from a representative frame, capturing salient visual context. These features are then integrated with structured prompts to guide a large language model in synthesizing natural dialogue. To ensure contextual and emotional consistency across scenes, we introduce a *Recursive Narrative Bank*—a speaker-aware, temporally structured memory that recursively accumulates each character's dialogue history. Inspired by Script Theory in cognitive psychology, this design enables characters to speak in ways that reflect their evolving goals, social context, and narrative roles throughout the story. Finally, we render each utterance as expressive, character-conditioned speech, resulting in fully-voiced, multimodal video narratives. Our training-free framework generalizes across diverse story settings—from fantasy adventures to slice-of-life episodes—offering a scalable solution for coherent, character-grounded audiovisual storytelling.

## 1 Introduction

Multimodal storytelling Lin & Chen (2024); Yang et al. (2024a); Zang et al. (2024); Arif et al. (2024); Bae et al. (2023); Sohn et al. (2024); Zhang et al. (2024b); Xu et al. (2025) aims to generate rich, coherent narratives by combining visual, textual, and auditory modalities. While recent advances in storytelling video generation have enabled systems such as Text2Story Kang et al. (2025) to synthesize temporally coherent scenes from structured prompts, these pipelines predominantly focus on visual consistency, often overlooking the linguistic expressivity essential to compelling narratives. In particular, the generation of character-driven dialogue remains a largely unaddressed challenge. Although scenes may accurately depict spatial configurations and motion transitions, the absence of character speech significantly limits the emotional depth and narrative realism of the resulting stories.

One of the core difficulties in multimodal storytelling is the transformation of high-level action prompts into dialogue that feels natural, context-aware, and consistent with a character's persona. Consider, for example, the following pair of scene prompts: *"The sun begins to set over the Pacific Ocean"* (nearby Golden Gate) followed by *"Shrek and Donkey are standing"*. A fitting line of dialogue in this case might be Donkey exclaiming, *"(Sky is) Prettier than Fiona!"*—a humorous and character-consistent utterance. In a later scene, one of the prompts described as *"Shrek and Donkey watch the Golden Gate Bridge in the evening"*, Shrek responds with *"Nothing tops Fiona's beauty even the Golden Gate!"*, demonstrating how dialogue can reflect continuity, rivalry, and character dynamics over time. Mapping these prompts to appropriate utterances requires not just scene understanding but also awareness of prior exchanges and character relationships. Figure 1 illustrates this exchange, showing how our system generates emotionally grounded dialogue that evolves across scenes. Captioning models or template-based systems fall short in capturing such narrative nuance, as they are not designed to model evolving character interactions or dialogue history. To address this gap, we present a pipeline that transforms structured scene prompts into coherent, character-driven dialogue and speech. Our approach assumes a pair of text prompts per

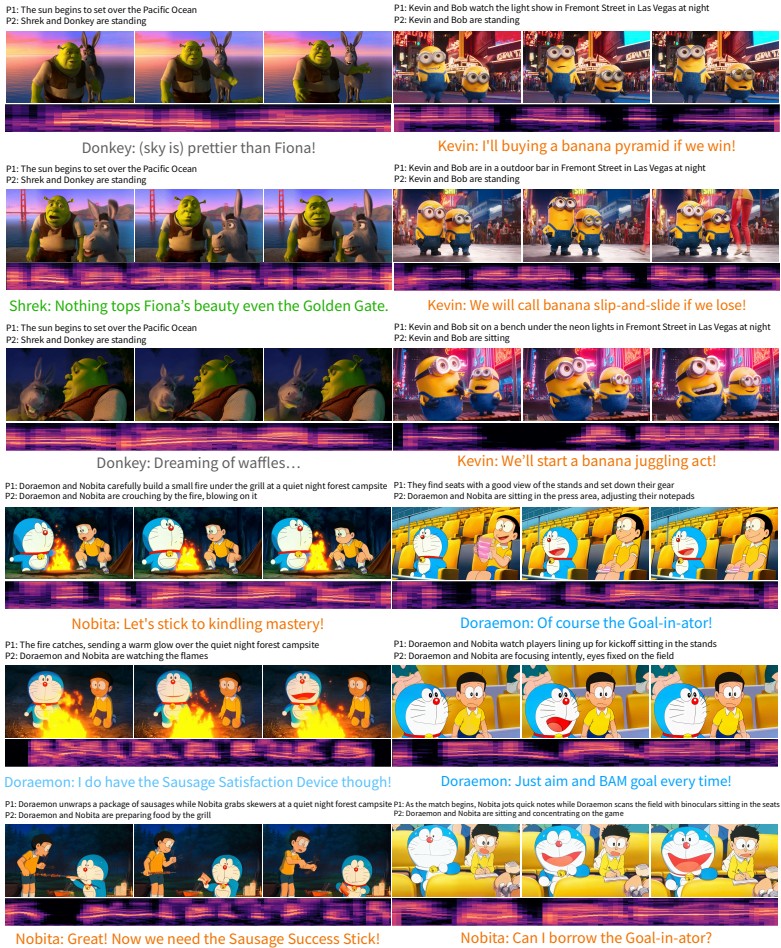

Figure 1: **Storytelling across Thematic Settings.** We showcase multimodal storytelling across four distinct settings using animated characters. **(a) Urban Exploration in San Francisco** (Top, L) Shrek and Donkey reflect on the beauty of the Pacific sunset and the Golden Gate, blending humorous banter with visual splendor. **(b) Urban Exploration in Las Vegas** (Top, R): Kevin and Bob (Minions) navigate the vibrant nightlife of Fremont Street with playful bets, envisioning banana pyramids and juggling acts under neon lights. **(c) Outdoor Cooking Show** (Bottom, L): Doraemon and Nobita build a campfire and engage in a light-hearted cooking segment, featuring whimsical gadgets like the "Sausage Satisfaction Device." **(d) Sports Reporting** (Bottom, R): Doraemon and Nobita become animated sports commentators, highlighting game moments with imaginative tools, like the "Goal-in-ator."

scene: the first defines the setting, and the second describes a character's action. While the visual content is generated using story generation models such as Text2Story Kang et al. (2025), we focus on producing dynamic and contextually appropriate dialogue. We first apply a pretrained vision-language encoder to extract a high-level semantic feature from the scene's representative keyframe, capturing salient visual context. This feature, together with the scene prompts and accumulated dialogue history, is used to guide a large language model in generating natural, persona-consistent utterances.

We introduce a *Recursive Narrative Bank (RNB)*, a speaker-aware and cognitively grounded memory structure that recursively accumulates and organizes dialogue across scenes. Unlike stateless GPT APIs, which require users to manually concatenate past messages or even chat-based interfaces, such as ChatGPT, which retains dialogue history without structuring it by speaker or temporal context, RNB maintains **role-conditioned, temporally structured memory for each character, allowing the model to generate contextually appropriate and emotionally consistent dialogue over time.** By conditioning the language model on this recursively structured memory — alongside current prompts and scene-level visual inputs — RNB enables characters to naturally refer to prior events, express evolving emotions, and maintain narrative continuity. This mirrors human conversational behavior, where utterances are shaped not by rote retrieval of prior sentences but by social context, emotional states, and situational scripts. Our design draws inspiration from **Script The-**

**ory Schank & Abelson (2013); Bower et al. (1979); Wilensky (1983) in cognitive psychology**, which posits that people use structured sequences of roles and events to interpret and produce appropriate behavior in social scenarios. RNB operationalizes this theory within a language model prompt design, enabling training-free, long-form, multimodal dialogue generation that remains coherent across scenes, grounded in visual context, and consistent with each character's persona.

To further enhance immersion, we render the generated dialogue as expressive, character-consistent speech using a reference-driven voice synthesis approach. Short audio clips paired with aligned transcripts are used to implicitly capture character-specific vocal traits and prosody, producing emotionally rich and narratively grounded speech. This integration of vision, language, and voice enables lifelike multimodal storytelling without the need for additional model training. Unlike previous approaches that rely on curated scripts or fixed templates, our framework is training-free, modular, and adaptable. It supports diverse characters, scenes, and storytelling styles while ensuring expressive, multimodal coherence. We demonstrate our system through a variety of examples, such as Shrek and Donkey's comical journey through diverse scenes in San Francisco, showcasing its ability to generate engaging dialogue, consistent speech, and cohesive story progression. In summary, we introduce a scalable, learning-based system for multimodal story generation that tightly integrates image captioning, large language models, and speech synthesis. Our method bridges the gap between layout-driven video generation and character-driven storytelling, providing a foundation for interactive story authoring and grounded dialogue generation in AI-empowered media.

**Our key contributions are summarized as follows:**

1. **Scene-to-Dialogue Transformation from Structured Prompts.** We propose a modular pipeline that transforms high-level scene and action prompts into fluent, persona-consistent character dialogue. By conditioning a large language model on both image-grounded captions and structured prompts, our system bridges the gap between action description and natural utterance synthesis.

2. **Recursive Narrative Bank for Context-Aware Dialogue.** We introduce a cognitively grounded, speaker-aware memory mechanism that recursively accumulates role-conditioned dialogue across scenes. By integrating this structure into the prompt design, our method enables contextually appropriate, emotionally consistent, and visually grounded dialogue generation—allowing characters to naturally reference past events and maintain narrative continuity over time.

3. **Expressive Speech Rendering for Multimodal Storytelling.** We synthesize character-specific speech aligned with generated dialogue using a reference-driven voice rendering approach. By providing short audio samples and aligned transcripts for each character, our method implicitly captures speaker identity and prosody without requiring manual embeddings or model fine-tuning. This enables emotionally expressive, character-consistent voice output that enhances immersion across diverse narrative settings.

4. Through comprehensive quantitative and qualitative evaluation, we demonstrate the effectiveness of our framework. Our method achieves a **+71.9% improvement in BERTScore** and **+45.0% in BLEU** over ablated variants without visual grounding, and **reduces DTW by 67.3%** when compared to speech generation without speaker conditioning. In a human subject study across five settings, overall our system was **preferred in 87.2% of cases** over other alternatives, with users citing natural dialogue and character-consistent tone as the key factors of user preference for our system.

## 2 RELATED WORK

### 2.1 STORYTELLING VIDEO GENERATION

Storytelling video generation has evolved from script-to-video pipelines into multimodal systems capable of handling complex narratives. Early works Huang et al. (2016); Pan et al. (2024); Liu et al. (2024) focus on generating coherent narratives or image sequences from multi-sentence descriptions or photo albums. While these approaches provide important foundations for visual storytelling, they remain limited to static images and do not address dialogue, character modeling, or speech synthesis. Recent methods such as VideoDirectorGPT Lin et al. (2023) and Vlogger Zhuang et al. (2024) decompose scripts using large language models (LLMs) to guide scene-wise synthesis. Animate-A-Story He et al. (2023) enhances this by retrieving depth-conditioned references to improve motion control. DreamStory He et al. (2024) and MovieDreamer Zhao et al. (2024)

adopt a two-stage approach—generating keyframes with diffusion models and animating them via image-to-video models. However, sparse keyframes can cause unnatural transitions. StoryDiffusion Zhou et al. (2025) introduces Consistent Self-Attention and a Semantic Motion Predictor to improve frame consistency but still struggles with complex scenes and requires model training. DreamRunner Wang et al. (2024b) advances motion control using retrieval-augmented priors and spatial-temporal 3D attention, enhancing adaptability but still lacking seamless scene-to-scene coherence. TTT-Video-DiT Dalal et al. (2025) extends video length through adaptive memory and test-time training, producing minute-long videos from storyboard inputs. However, its end-to-end nature results in limited scene-level controllability, making it difficult to trace individual prompts or evaluate scene-to-dialogue correspondence. Moreover, it lacks mechanisms for dialogue modeling, speaker tracking, or audio generation. Text2Story Kang et al. (2025) proposes a structured approach for long-form storytelling from prompt sequences, blending short video clips using prompt-space mixing and latent blending. While it improves visual coherence without training, it lacks character dialogue and voice. To address this gap, our method enhances narrative coherence by introducing character-grounded dialogue and expressive speech synthesis, enabling more engaging storytelling beyond visual scenes.

## 2.2 MULTIMODAL STORYTELLING

Recent advances in generative models have enabled multimodal storytelling systems that integrate text, visuals, and sound to improve narrative expressiveness and user engagement. Improving Visual Storytelling with Multimodal LLMs Lin & Chen (2024) and SEED-Story Yang et al. (2024a) leverage vision-language models to generate coherent visual narratives. However, both focus on image-text generation and do not support spoken dialogue or character-specific utterances. SEED-Story is further limited by low resolution (480×768), unlike our 848×480 video generation with synchronized speech. Several methods explore audio integration. LLaMS Zang et al. (2024) enhances narrative expressiveness with commonsense reasoning but generates narrator-style stories. Art of Storytelling Arif et al. (2024) supports co-created narration with TTS but lacks character-specific identity and dialogue. Sound of Story (SoS) Bae et al. (2023) provides background audio paired with image-text stories, but without speech or utterances. StoryAgent Sohn et al. (2024) orchestrates multimodal narratives using LLMs and asset generators, but does not handle scene-specific dialogue or speaker identity. Dialogue Director Zhang et al. (2024b) translates scripts into visual sequences with diffusion models, improving visual controllability without spoken output. MM-StoryAgent Xu et al. (2025) builds narrated storybooks via multi-agent planning, offering music and narration but not character-grounded speech. While many of these methods Lin & Chen (2024); Yang et al. (2024a); Zang et al. (2024); Arif et al. (2024); Bae et al. (2023); Sohn et al. (2024); Zhang et al. (2024b); Xu et al. (2025), including LLaMS, Art of Storytelling, Improving Visual Storytelling with Multimodal LLMs, StoryAgent, Dialogue Director, and MM-StoryAgent, adopt GPT-based or multimodal generation architectures, and SEED-Story also leverages a large language model, **none of them achieve the level of expressiveness, character grounding, and speech synchronization that our system demonstrates.** Our architecture is not a mere application of GPT; rather, it is a purpose-built pipeline that explicitly grounds behavior prompts into utterances and generates high-quality, character-consistent speech. **The performance gains observed in our results stem not from the backbone model alone, but from our novel integration of vision-language grounding, dialogue modeling, and speech synthesis.** In contrast to prior work, we focus on generating expressive, character-driven utterances from scene and behavior prompts, and converting them into natural speech via a conversational speech model. Our approach bridges the gap between visual storytelling and character-based dialogue generation, enabling immersive narrative experiences with multimodal expressiveness grounded in both image and context.

## 2.3 TEXT-TO-VIDEO DIFFUSION, DIALOGUE GENERATION FROM VISUAL CONTEXT & TEXT-TO-SPEECH

A wide range of studies have explored text-to-video generation Bar-Tal et al. (2024); Chen et al. (2023a); Fei et al. (2024); Girdhar et al. (2023); Khachatryan et al. (2023); Qing et al. (2024); Singer et al. (2022); Wang et al. (2023b); Weng et al. (2024); Zhang et al. (2024a; 2023); Henschel et al. (2024); Qiu et al. (2023); OpenAI (2024); Blattmann et al. (2023b); Ge et al. (2023); Wang et al. (2024a); Yin et al. (2023); He et al. (2022); Blattmann et al. (2023a); Wang et al. (2023a); Chen et al. (2023b); Oh et al. (2024); Villegas et al. (2022); Polyak et al. (2025); Sharma et al. (2024); Veo-Team et al. (2024). Mochi AI (2024a) and CogVideoX Yang et al. (2024b) have advanced temporal control and resolution, yet struggle with coherent multi-shot storytelling and fine-grained scene grounding, which limits their effectiveness in narrative-centered applications. Dialogue generation

has traditionally relied on full scripts or high-level summaries. In contrast, we generate natural utterances directly from action-level prompts. Captioning and summarization models fall short in modeling interaction or persona. While various vision-language models Li et al. (2022; 2023a); Radford et al. (2021); Alayrac et al. (2022); Zhu et al. (2023); Dai et al. (2023); Wang et al. (2022a); Chen et al. (2022); Kim et al. (2021); Chen et al. (2020); Li et al. (2021); Wang et al. (2022b); Yu et al. (2022); Peng et al. (2023); Fu et al. (2021); Li et al. (2023b); Luo et al. (2023) demonstrate strong cross-modal reasoning, we adopt BLIP Li et al. (2022) for its generative compatibility and structured prompt conditioning, enabling visually grounded, persona-consistent dialogue generation. For speech synthesis, models like Tacotron Wang et al. (2017) and Bark AI (2023) produce fluent speech but lack character specificity. SesameAILabs' CSM AI (2024b) enables expressive, identity-aware voice synthesis through reference audio and RVQ-based tokenization. Inspired by this, we employ a reference-driven approach that preserves character identity and emotional nuance, aligning expressive voice with generated dialogue and visual context.

# 3 METHOD

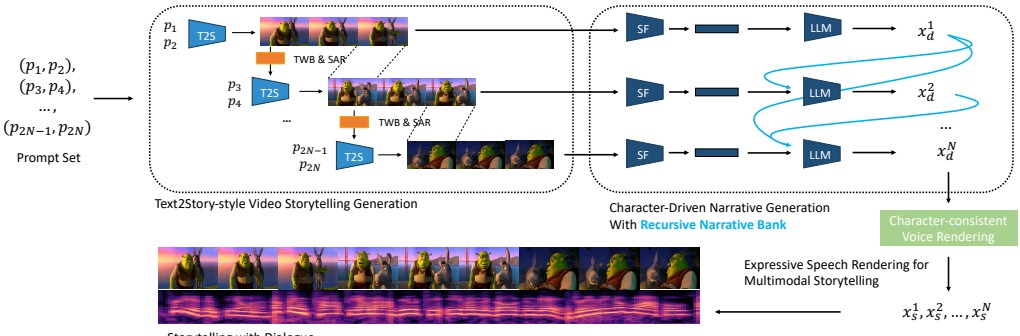

Figure 2: **Overview of our proposed multimodal storytelling framework.** Given a sequence of paired prompts $(p_1, p_2), (p_3, p_4), \ldots, (p_{2N-1}, p_{2N})$, our system generates coherent video scenes, natural dialogue, and expressive speech. A story generation model (e.g., Text2Story (T2S)) synthesizes short video clips for each prompt pair, which are blended using Time-Weighted Blending (TWB) and refined with Semantic Action Representation (SAR) to ensure temporal coherence. For each scene, we extract a representative frame using `SceneFeat` (SF) to obtain a semantic visual feature. This, along with the current prompt pair and a Recursive Narrative Bank of prior dialogue, is used to prompt a large language model for generating character-consistent dialogue. Finally, the generated utterance is converted into speech with reference-driven conditioning to produce high-fidelity, emotionally expressive voice output. The result is a fully voiced, visually grounded, and narratively coherent story video.

Our goal is to generate character-grounded narrative dialogue and corresponding speech from structured video scene prompts. Each scene is described by a pair of prompts: one describing the setting $(p_1)$ and the other specifying a character's action $(p_2)$. This two-prompt formulation, inspired by the design of Text2Story Kang et al. (2025), enables precise scene grounding while allowing fine-grained control over character behavior and interactions. The overall process is a composition of three conditional mappings:

$$(x_v, x_d, x_s) = \mathcal{F}(p_1, p_2, H), \tag{1}$$

where $x_v$ is the generated visual scene, $x_d$ is the character dialogue, $x_s$ is the synthesized speech, and $H$ is the dialogue history from prior scenes, maintained through a *Recursive Narrative Bank*. The mapping $\mathcal{F}$ consists of three stages: **Scene Visualization**, **Dialogue Generation**, and **Speech Synthesis**.

## 3.1 SCENE VISUALIZATION

To generate visual grounding, we assume a story generation model (e.g., Text2Story) that takes $(p_1, p_2)$ as input and synthesizes a short video clip $x_v$. We uniformly sample $K$ keyframes $I_1, I_2, ..., I_K$ from $x_v$ such that:

$$I_k = \text{SampleFrame}(x_v, t_k), \quad k \in [1, K], \tag{2}$$

where $t_k$ are evenly spaced timestamps over the video duration. For efficiency, we use the middle frame $I = I_{\lfloor K/2 \rfloor}$ as the representative scene image for subsequent dialogue generation.

## 3.2 DIALOGUE GENERATION

This stage generates character-consistent dialogue conditioned on both the current scene and prior narrative context. Let $p = p_1 + `` . \quad '' + p_2$ denote the full scene prompt. For each representative frame $I$, we extract a high-level semantic representation $c$ using a pretrained vision-language encoder:

$$c = \texttt{SceneFeat}(I), \tag{3}$$

where `SceneFeat` is implemented as BLIP Li et al. (2022). The resulting embedding $c$ is decoded into a natural language caption, which is inserted into the structured prompt to provide semantically grounded visual context compatible with standard LLM APIs.

To ensure narrative continuity, we introduce a **Recursive Narrative Bank (RNB)** $\mathcal{H}$, a dynamic buffer of prior utterances. At scene step $t$, we define:

$$H_t = \{x_d^{(t-1)}, x_d^{(t-2)}, \dots, x_d^{(t-N)}\}, \tag{4}$$

where $x_d^{(i)}$ denotes the generated dialogue at step $i$, and $N$ is a fixed window size. We define a Recursive Narrative Bank as a memory structure where each scene's generation is informed by the full history of previous outputs. ($N$ = all scenes) This recursive setup ensures that contextual information accumulates throughout the narrative, allowing characters and events to evolve consistently across scenes. Given $c$, $p$, and $H_t$, we generate dialogue $x_d$ as:

$$x_d = \texttt{LLM}(c, p, H_t), \tag{5}$$

where `LLM` is a pretrained large language model, conditioned through a structured prompt template that integrates both visual semantics and temporally bounded dialogue memory. Rather than using flat concatenation, we encode the current scene context via a three-part narrative control signal:

$$\texttt{Input} = \texttt{NarrativePrompt}(c, p, H_t) = [\texttt{Scene}]\, p \quad \| \, [\texttt{Image}]\, c \, \| \, [\texttt{DialogueMemory}]\, H_t \tag{6}$$

where each segment is demarcated by explicit semantic tags. The `[Scene]` and `[Image]` sections capture the spatial and semantic grounding of the current scene, while the `[DialogueMemory]` section represents a bounded rolling window of prior utterances drawn from the Recursive Narrative Bank. By enforcing this structured decomposition, the model learns to interpret prior exchanges not merely as text, but as dynamic dialogue state. This formulation enables controlled narrative flow by preserving coherence across temporally adjacent scenes while adapting to new visual contexts. It also supports scalable story generation by decoupling dialogue planning from low-level text generation, allowing fine-grained modulation of persona, mood, and thematic consistency. More details about RNB are available in the Appendix A.5.

## 3.3 SPEECH SYNTHESIS

Once dialogue text $x_d$ is generated, we render natural-sounding speech $x_s$ through a reference-driven voice synthesis approach tailored for character expressivity. Rather than relying on predefined speaker embeddings or fine-tuning, our method conditions speech generation on short audio clips paired with aligned transcripts for each character. These reference segments—extracted from publicly available character voices (e.g., Shrek and Donkey)—enable the model to implicitly learn speaker-specific prosody and vocal traits.

Given the target dialogue and character identity, the system synthesizes expressive audio by considering both the current utterance and a small context window of preceding dialogue turns. This design ensures temporal consistency in vocal delivery while maintaining emotional nuance and character fidelity across scenes. The resulting speech closely reflects each character's personality, enhancing immersion and narrative realism in the final video output. More details about speech synthesis are available in the Appendix A.6.

## 4 EXPERIMENTS AND RESULTS

### 4.1 BENCHMARKING DATASETS

We evaluate our method on a variety of **narrative scenarios** to demonstrate its broad applicability. The scene settings include:

- **Urban exploration in San Francisco and Las Vegas:** We leverage the Text2Story dataset, which provides structured scene and action prompts. These include character journeys through landmarks, such as the Golden Gate Bridge, San Francisco cable cars, and the Las Vegas hotel/casino.

- **Cooking show:** We construct a cooking show scenario set around a forest campfire, where characters go through the step-by-step process of making hot dogs. This instructional setting highlights our system's ability to generate semantically grounded and temporally coherent dialogue.

- **Sports reporting:** We also include a sports reporting scenario in which characters attend a soccer match as commentators, offering observations and analysis. This domain tests our model's capacity for context-aware and role-consistent dialogue.

Across all scenarios, we use multiple character pairs with distinct personalities (first creation date): (a) Shrek & Donkey (2001), (b) Doraemon & Nobita (1969), (c) Tom & Jerry (1940), and (d) Minions Kevin & Bob (2015) — ensuring that our pipeline supports diverse storytelling contexts and expressive styles. We explicitly choose cartoon characters introduced across different eras to demonstrate the temporal and cultural range of our storytelling capability, encompassing both classic and modern animated narratives. Cartoon voices are particularly suitable for our benchmark because (i) short audio excerpts are readily available from official YouTube channels, and (ii) their speech is typically clearer, more stylized, and more rhythmic than that of real human speakers, making them easier to model and synthesize in a controlled setting. To synthesize character-consistent speech, we use short audio excerpts from publicly available YouTube videos released by official sources, strictly for academic, non-commercial research purposes. These materials are used in accordance with fair use principles and do not involve redistribution of copyrighted content. Importantly, we do not replicate full voice likenesses; our generated voices are expressive approximations rather than imitations of real actors, minimizing ethical concerns. Each story contains 11 to 13 scene-level plots, and every plot is described using a pair of prompts: one for the setting and one for the character's action. In total, our dataset comprises 408 structured inputs used for narrative generation. Details are available in Appendix A.8.

## 4.2 IMPLEMENTATION DETAILS

We implement our pipeline using HuggingFace Transformers and SesameAILabs' CSM API AI (2024b). Visual scenes are generated using a Text2Story model Kang et al. (2025). BLIP Li et al. (2022) (image captioning base) is used for frame-level description. For dialogue generation, GPT-4o Achiam et al. (2023) is used in chat-completion mode with a history window size $N = $ all in the Recursive Narrative Bank, enabling long-range coherence across scenes. Since each utterance is short (under 100 tokens), full history fits within context limits and helps preserve character consistency without truncation. Speech synthesis is performed using the CSM model in inference mode, with speaker embeddings mapped to canonical character voices. The pipeline is fully modular and supports easy substitution of individual components for domain-specific adaptation or fine-tuned extensions.

## 4.3 QUALITATIVE RESULTS

We present qualitative examples showcasing the effectiveness of our character-driven narrative generation framework. In Figure 1, we illustrate how our system generates expressive, context-aware dialogue across diverse thematic settings. The examples feature familiar animated characters engaging in urban exploration, outdoor cooking, and sports commentary, each accompanied by distinctive vocal tone and narrative alignment. In a San Francisco sunset scene, Shrek and Donkey overlook the Pacific Ocean as Donkey playfully remarks that the sky is prettier than Princess Fiona. His teasing tone is lighthearted, clearly meant as a joke, but Shrek reacts with mild irritation—defending Fiona's beauty with a sharp but composed response that "Nothing tops Fiona's beauty even the Golden Gate." Capturing this kind of nuanced emotional interplay—where one character jokes and the other responds with restrained frustration rather than overt anger, both video and narration—requires fine-grained control over dialogue tone and character intent. Our model preserves each character's personality and adjusts their reactions in accordance with subtle shifts in emotional context, demonstrating its strength in character-consistent narrative generation. In contrast, the Las Vegas sequence highlights Kevin and Bob's slapstick banter as they react to the neon-drenched cityscape. Their dialogue is filled with playful bets about building banana pyramids or launching a juggling act, reflecting both the visual absurdity of their surroundings and their food-driven motivation. The model successfully sustains their energetic tone and synchronizes their speech with visual cues like

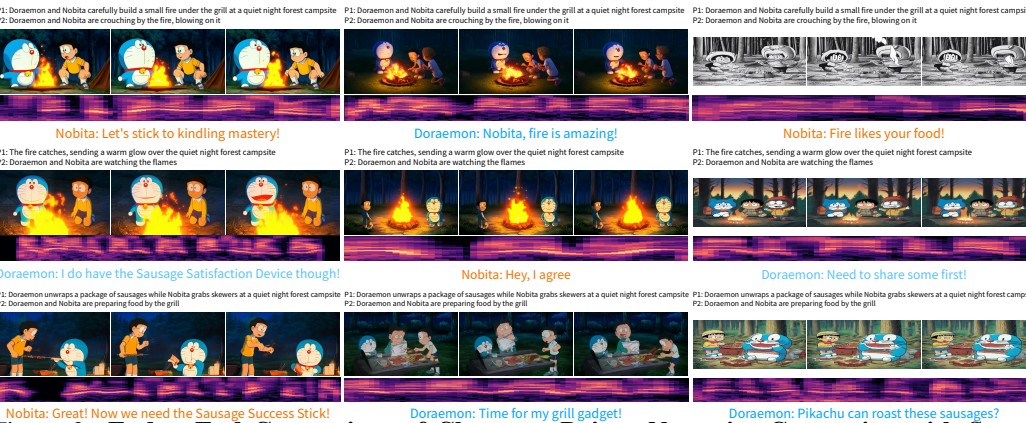

Figure 3: **End-to-End Comparison of Character-Driven Narrative Generation with Speech Rendering.** We qualitatively compare three systems—Ours (left), Mochi + Speech Rendering (middle), and Vlogger + Speech Rendering (right)—in generating multimodal dialogue sequences from structured prompts. Each sequence includes sampled frames, generated dialogue, and corresponding audio spectrograms. While all systems receive identical prompt inputs, our method demonstrates superior narrative grounding by selecting more semantically appropriate scenes for each prompt pair. This results in dialogue that is not only more fluent and expressive, but also better aligned with the visual context. The comparison highlights that scene selection plays a critical role in enabling coherent and character-consistent narrative generation.

pointing gestures and crowd reactions. In the outdoor cooking scene, Doraemon and Nobita huddle around a campfire as they attempt to master kindling and prepare skewers using whimsical gadgets. The generated dialogue includes Nobita's earnest attempts at fire-building and Doraemon's matter-of-fact responses about having the "Sausage Satisfaction Device", showcasing a cozy and supportive dynamic. Here, the model adapts to a slower pace and softer emotional tone while preserving character consistency. A direct comparison of this scene against baseline methods is shown in Figure 3, where our approach produces more contextually grounded and emotionally expressive dialogue, supported by appropriate scene selection and synchronized speech rendering. Later, during a sports reporting, the same characters appear in a stadium, enthusiastically watching the game and providing commentary. Nobita jots down notes while Doraemon scans the field, offering confident advice on how to score with the "Goal-in-ator". Despite the domain shift, their personalities remain intact—Nobita's curiosity and Doraemon's problem-solving confidence are clearly preserved in both the text and synthesized speech. Across all cases, our system effectively combines visual understanding, prompt conditioning, and prior dialogue context to produce coherent, engaging, and character-consistent narratives. The resulting speech not only reflects each character's vocal identity but also enhances immersion through appropriate emotional expression and dialogue timing. Full video examples and audio clips are available in the supplementary materials.

## 4.4 QUANTITATIVE EVALUATION

We quantitatively evaluate the effectiveness of our character-driven multimodal storytelling framework using four automated metrics: BERTScore, BLEU, CLIPScore, and Dynamic Time Warping (DTW). Each metric targets a specific aspect of the generation pipeline, enabling comprehensive assessment of linguistic, semantic, multimodal, and acoustic quality without requiring human annotation. To align with established evaluation standards in story generation research, we follow the evaluation setup introduced in Text2Story Kang et al. (2025). Unless otherwise specified, all evaluation settings follow this prior work. More details about quantitative results are available in Appendix A.4.

## 4.5 ABLATION STUDY

To understand the contribution of each module in our multimodal storytelling framework, we conduct ablation studies across key components and evaluate performance using BERTScore, BLEU, CLIPScore, and Dynamic Time Warping (DTW). These metrics provide a comprehensive view of linguistic fidelity, visual grounding, and prosodic expressivity. More details about full qualitative results and quantitative results are available in Appendix A.4.

## 4.6 USERS EVALUATION

To evaluate the quality, coherence, and emotional expressiveness of our character-driven video generation framework, we conducted a human subject study with 15 participants. The study aimed to assess which video-audio combinations were perceived as most vivid, natural, and immersive by human viewers, without revealing the underlying model names. Figure 4 shows the aggregated preference counts for each video condition across the five experimental settings. Our model (Video A) was selected in 197 out of 225 total responses, corresponding to a preference rate of **87.2%**. This

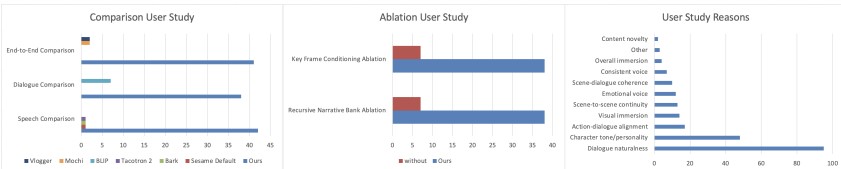

Figure 4: **Human Subjects Evaluation Results.** (Left) Model comparison results across three conditions (Speech, Dialogue, End-to-End). Our model (dark blue) was preferred in the vast majority of cases, outperforming all baselines including Tacotron 2, Bark, BLIP, Mochi, and Vlogger. (Center) Ablation results for Recursive Narrative Bank and Key Frame Conditioning. Our full model again dominates, indicating the importance of these components for coherence and alignment. (Right) Participant justifications for their preferred choices. Most frequently cited reasons include *"dialogue felt natural and matched the scene"* and *"character tone and personality"*, reflecting the narrative quality and expressiveness of our generation pipeline.

| Method | BERTScore ↑ | BLEU ↑ | CLIPScore ↑ | DTW ↓ |
|---|---|---|---|---|
| **Ours (Full Pipeline)** | **0.0674** | **1.8726** | **27.4822** | **16.2412** |
| *Dialogue Generation* | | | | |
| BLIP | 0.0674 | 1.8726 | 27.4822 | 17.0377 |
| *Speech Generation & Speaker Conditioning* | | | | |
| Speech Rendering w/o character embedding | 0.0674 | 1.8726 | 27.4822 | 49.6302 |
| Bark | 0.0674 | 1.8726 | 27.4822 | 16.4668 |
| Tacotron 2 | 0.0674 | 1.8726 | 27.4822 | 16.4484 |
| *End-to-end Comparison* | | | | |
| Mochi + Speech Rendering | 0.0092 | 0.1272 | 26.6107 | 17.7431 |
| Vlogger + Speech Rendering | 0.0094 | 0.2648 | 26.1703 | 18.6999 |
| *Recursive Narrative Bank* | | | | |
| w/o RNB | 0.0094 | 0.5268 | 27.2259 | 16.5604 |
| *Key Frame Image Conditioning* | | | | |
| w/o Conditioning | 0.0392 | 1.2914 | 25.3061 | 22.4406 |

Table 1: **Quantitative evaluation of dialogue and speech-integrated video storytelling.** Higher BERTScore, BLEU, and CLIPScore indicate better alignment with text and ground-truth captions, while lower DTW reflects smoother temporal consistency in speech and narrative flow.

strong preference was consistent across all experiment types. Notably, in the Speech Generation comparison, our method received 42 out of 45 votes (**93.3%**), while in the Dialogue Generation setting, it received 38 out of 45 votes (**84.4%**). Details are available in Appendix A.3. In addition to quantitative preference, the free-text and multiple-choice justifications revealed common themes:

- Participants often selected our model because the character's tone, emotions, and vocal style felt "consistent" and "expressive".
- Users favored versions with the Recursive Narrative Bank for "story flow" and "continuity between scenes".
- Keyframe-based visual grounding improved alignment between scene and utterance, contributing to perceived coherence.

## 5 CONCLUSION

In this paper, we presented a modular framework for character-driven narrative generation in scene-based video synthesis. Our system augments structured prompt-based video generation with expressive, context-aware dialogue and speech, enabling visually and auditorily grounded storytelling experiences. By integrating vision-language models, large language models, and speaker-conditioned text-to-speech synthesis, our method captures both visual semantics and temporal narrative flow. We introduced the Recursive Narrative Bank, a novel dialogue memory mechanism that enables temporally coherent, character-consistent utterances across scenes. Through our experiments, we demonstrate that the proposed pipeline enhances character believability, enriches narrative progression, and adds emotional resonance to AI-generated videos without requiring model retraining or domain-specific fine-tuning. **Limitations and Future Directions:** Despite the strengths of our system, several limitations remain. The reliance on a single representative frame for captioning may omit important dynamic context that occurs in other parts of the video. Moreover, speech synthesis using CSM, while expressive, is restricted to mono-channel output and limited token lengths, which may impact scalability in longer scenes or multi-character interactions. Future work could aim to explore multimodal memory extensions that encode not just past dialogue but also visual and emotional state evolution. Additionally, integrating user-controlled editing and feedback mechanisms would enable more interactive and personalized storytelling experiences, pushing the boundary of GenAI-based content creation in narrative media.

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

## A  Supplementary Material

### A.1  Results - Readme: Before Watching Compressed Videos

Please refer to the supplementary video for more results.

---
**README: BEFORE WATCHING COMPRESSED VIDEOS**

Dear Reviewers,
Due to the **100MB submission limit** for ICLR 2026, **we had to significantly compress the supplementary files.** The original file size for the Main Results video was 179.3MB, and the Comparison video was 116MB. Please understand that the **compression version of videos may introduce visible artifacts, and we are happy to provide the original high-quality videos during the review process** if needed and approved by the conference.
Sincerely,
The Authors

---

### A.2  Ethics Statement

---
**Ethics Statement**

All audio samples used in this work are limited to short excerpts for non-commercial, academic research purposes. No full scenes or monetized content are used, and speaker identities are simulated for character voice generation. We adhere to fair use guidelines and will release only anonymized and text-aligned metadata in compliance with copyright standards. We further clarify that all cartoon characters depicted in this research (Shrek & Donkey, Doraemon & Nobita, Tom & Jerry, and Minions Kevin & Bob) are used solely as fictional references to evaluate storytelling and voice synthesis capabilities. These characters are not used for profit, distribution, or endorsement, and their inclusion is intended for academic demonstration under fair use. Generated audio and visuals are produced synthetically and do not involve the use of any original footage or proprietary media. Our focus remains on advancing generative modeling techniques for educational and research purposes within responsible AI practices.

---

#### A.2.1  License for existing assets

---
**License for existing assets**

We utilize several publicly available pretrained models in our framework, each of which is used in accordance with its respective open-source license. Specifically, the Mochi-1 video generation model is distributed under the Apache 2.0 License (`https://github.com/genmoai/mochi`), and is used for scene-level visual synthesis. The Sesame Conversational Speech Model (CSM), used for character-specific expressive speech generation, is also distributed under the Apache 2.0 License (`https://huggingface.co/sesame/csm-1b`). In addition, the BLIP model, which is distributed under the Creative Commons Attribution 4.0 International (CC BY 4.0) License (`https://huggingface.co/Salesforce/blip-image-captioning-base`). All models are used without modification and solely for academic, non-commercial research purposes. We ensure proper attribution and full compliance with each model's licensing terms.

---

### A.3  User Study

**Survey Setup and Interface.** Participants were presented with 15 comparison questions, each showcasing 2–4 short AI-generated video clips (labeled only as "Video A", "Video B", etc. to ensure fairness). Each video was accompanied by synthesized character speech and dialogue. The clips varied across different experimental settings including speech synthesis methods, dialogue generation approaches, and ablation configurations.

After watching each video set, participants were asked to select the most vivid and engaging clip and explain their reasoning. They could either choose from a list of predefined qualitative factors or write open-ended comments. The predefined options included criteria such as:

- *Because the dialogue felt natural and matched the scene*
- *Because the character's tone and personality were well expressed*
- *Because the character's actions and dialogue were well aligned*
- *Because the background visuals or movements were natural and immersive*
- *Because the context between scenes was connected, making the story flow smoothly*
- *Because the voice conveyed rich emotions and suited the character*
- *Because the combination of scene and dialogue was intuitively understandable*
- *Because the character's voice was consistent and felt familiar*
- *Because overall, it felt immersive and vivid*
- *Because each scene provided new information without repetitive expressions*
- Participants could also submit free-form feedback if none of the listed items captured their impression.

**Conditions.**  The 15 questions were organized into five experimental categories:

- **Speech Generation & Speaker Conditioning (Q1–Q3):** Comparison across our full model, versions without character embeddings, and baseline systems like Bark and Tacotron.
- **Dialogue Generation (Q4–Q6):** Our system versus a BLIP-based caption-to-dialogue baseline.
- **End-to-End Comparison (Q7–Q9):** Full pipeline compared to prior works like Mochi and Vlogger.
- **Recursive Narrative Bank (Q10–Q12):** Ablation of our Recursive Narrative Bank (RNB).
- **Key Frame Image Conditioning (Q13–Q15):** Ablation of visual grounding in dialogue generation.

**Protocol.**  The survey took approximately 15–20 minutes. Participants were advised to use headphones for best audio quality but could participate via any device supporting video and audio. To respect IRB policy, no personal data was collected, and we do not compensate the participants. The study was reviewed and exempted by the Institutional Review Board (IRB).

**Statistical Summary.**  Out of **225** total responses, we observed consistently strong preference for our full model across all experiment categories. Rather than aggregating across all videos (which may appear in different subsets of questions), we report per-condition confidence intervals for Ours only. This avoids bias from unequal appearance of baseline systems. Table 2 summarizes the proportion of Ours selections and their 95% confidence intervals across the five experiment types. Also, this table further details per-condition confidence intervals for Ours. In all five experimental categories, our method was selected by a clear majority, with proportions ranging from 84.4% to 93.3%.

| Experiment | Ours Proportion | 95% CI Lower | 95% CI Upper | Total Votes |
|---|---|---|---|---|
| Speech Generation | 0.9333 | 0.8605 | 1.0000 | 45 |
| Dialogue Generation | 0.8444 | 0.7386 | 0.9503 | 45 |
| End-to-End Comparison | 0.9111 | 0.8280 | 0.9943 | 45 |
| Recursive Narrative Bank | 0.8444 | 0.7386 | 0.9503 | 45 |
| Key Frame Conditioning | 0.8444 | 0.7386 | 0.9503 | 45 |

Table 2: **Per-experiment condition preference for Video A with 95% confidence intervals.** Our method was selected by a clear majority, with proportion from 84.4% to 93.3% in all five categories.

**Reasoning Analysis.** Participants justified their selections using a list of qualitative reasons. Table 3 shows the distribution of all justifications. The most frequently selected reason was *"Because the dialogue felt natural and matched the scene"* (42.2%), followed by *"Character tone and personality"* (21.3%). Free-form responses that did not match predefined categories were grouped as "Other" (1.3%). Note that categories with fewer than 5 responses yielded wider confidence intervals with potentially negative lower bounds due to normal approximation, but these are clipped at zero in our interpretation.

| Reason Category | Proportion | 95% CI Lower | 95% CI Upper |
|---|---|---|---|
| Dialogue naturalness | 0.4222 | 0.3577 | 0.4868 |
| Character tone/personality | 0.2133 | 0.1598 | 0.2669 |
| Action-dialogue alignment | 0.0756 | 0.0410 | 0.1101 |
| Visual immersion | 0.0622 | 0.0307 | 0.0938 |
| Scene-to-scene continuity | 0.0578 | 0.0273 | 0.0883 |
| Emotional voice | 0.0533 | 0.0240 | 0.0827 |
| Scene-dialogue coherence | 0.0444 | 0.0175 | 0.0714 |
| Consistent voice | 0.0311 | 0.0084 | 0.0538 |
| Overall immersion | 0.0178 | 0.0005 | 0.0350 |
| Other | 0.0133 | 0.0000 | 0.0283 |
| Content novelty | 0.0089 | 0.0000 | 0.0212 |

Table 3: **Participant justifications for their preference with 95% CI:** the most frequently selected reason was "the dialogue felt natural and matching the scene".

## A.4 DETAILED QUANTITATIVE EVALUATION & ABLATION

*Due to a formatting error, please check the Datasets in Appendix 6, User Study in Appendix 3. We will revise that in the camera-ready version.*

**Detailed Quantitative Evaluation** To evaluate *dialogue quality*, we compute BERTScore and BLEU between generated utterances and their paired scene prompts and image-grounded captions. As summarized in Table 4, our full pipeline achieves the highest scores on both metrics (BERTScore: **0.0674 ± 0.057**, BLEU: **1.8726 ± 2.7237**), indicating strong semantic fidelity and fluency. Ablating visual grounding (i.e., "Without KeyFrame") degrades BERTScore to **0.0392 ± 0.1187** and BLEU to **0.8609 ± 1.1529**, with large variances suggesting unstable and inconsistent generation. Similarly, removing the Recursive Narrative Bank (RNB) drops BERTScore to **0.0094 ± 0.0353** and BLEU to **0.5268 ± 0.6288**, underscoring the RNB's role in linguistic and narrative coherence.

For *multimodal alignment*, we report CLIPScore between generated dialogue and keyframe images. Our model yields a CLIPScore of **27.4822 ± 3.4597**, outperforming alternative pipelines such as Vlogger+Speech (**26.1703 ± 4.0179**) and Mochi+Speech (**26.6107 ± 4.9015**). Notably, removing keyframe conditioning drops CLIPScore to **25.3061 ± 3.2293**, indicating weaker alignment with visual context.

To measure *speech expressivity*, we apply Dynamic Time Warping (DTW) over pitch contours to compare generated speech with reference character audio. Our system achieves the lowest DTW value (**16.2412 ± 5.7724**), indicating strong temporal and prosodic alignment. Ablating speaker rendering increases DTW substantially to **49.6302 ± 55.1206**, with the large standard deviation highlighting inconsistent prosody. Comparisons with Bark (**16.4668 ± 5.3581**) and Tacotron 2 (**16.4484 ± 0.9273**) show that our method achieves slightly better average performance while maintaining competitive expressivity with lower variance.

These results are summarized in Table 4. The consistent inclusion of standard deviations across all methods allows us to assess both performance and stability. Collectively, these metrics and their statistical spread confirm that each module—visual grounding, dialogue memory, and speech conditioning—contributes meaningfully and measurably to the multimodal storytelling quality.

**Detailed Ablation Study** We first examine the effect of keyframe-based visual grounding in dialogue generation. When image features are removed ("w/o Conditioning"), performance drops sharply across all metrics: BERTScore falls from 0.0674 to 0.0392, BLEU from 1.8726 to 1.2914,

| Method | BERTScore | BLEU | CLIP | DTW |
|---|---|---|---|---|
| **Ours** | **0.0674 ± 0.057** | **1.8726 ± 2.7237** | **27.4822 ± 3.4597** | **16.2412 ± 5.7724** |
| BLIP | 0.0674 ± 0.057 | 1.8726 ± 2.7237 | 27.4822 ± 3.4597 | 17.0377 ± 7.5249 |
| Speech w/o rendering | 0.0674 ± 0.057 | 1.8726 ± 2.7237 | 27.4822 ± 3.4597 | 49.6302 ± 55.1206 |
| Bark | 0.0674 ± 0.057 | 1.8726 ± 2.7237 | 27.4822 ± 3.4597 | 16.4668 ± 5.3581 |
| Tacotron 2 | 0.0674 ± 0.057 | 1.8726 ± 2.7237 | 27.4822 ± 3.4597 | 16.4484 ± 0.9273 |
| Mochi | 0.0092 ± 0.0474 | 0.1272 ± 0.1734 | 26.6107 ± 4.9015 | 17.7431 ± 4.8291 |
| Vlogger | 0.0094 ± 0.0842 | 0.2648 ± 0.6927 | 26.1703 ± 4.0179 | 18.6999 ± 7.7477 |
| Without RNB | 0.0094 ± 0.0353 | 0.5268 ± 0.6288 | 27.2259 ± 3.8112 | 16.5604 ± 5.1673 |
| Without KeyFrame | 0.0392 ± 0.1187 | 0.8609 ± 1.1529 | 25.3061 ± 3.2293 | 22.4406 ± 19.7112 |

Table 4: **Quantitative Evaluation with Mean ± Standard Deviation.** Our method achieves the best or competitive performance across linguistic, multimodal, and speech metrics, with standard deviations demonstrating consistent reliability.

P1: They find seats with a good view of the stands and set down their gear
P2: Doraemon and Nobita are sitting in the press area, adjusting their notepads

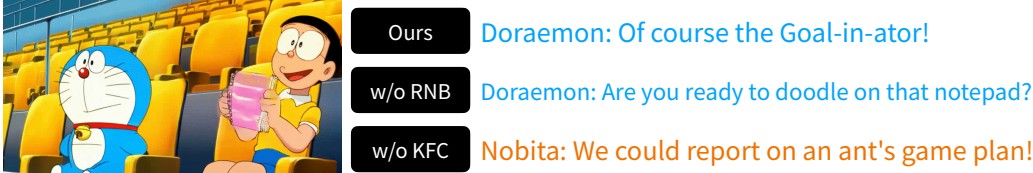

P1: Doraemon and Nobita watch players lining up for kickoff sitting in the stands
P2: Doraemon and Nobita are focusing intently, eyes fixed on the field

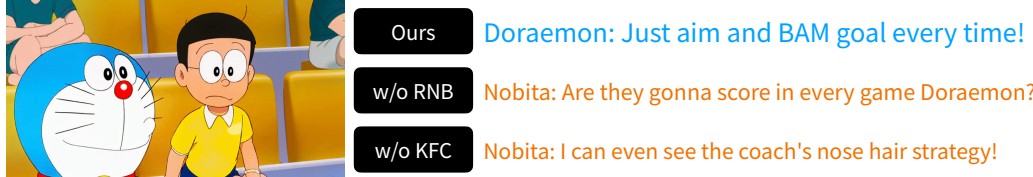

P1: As the match begins, Nobita jots quick notes while Doraemon scans the field with binoculars sitting in the seats
P2: Doraemon and Nobita are sitting and concentrating on the game

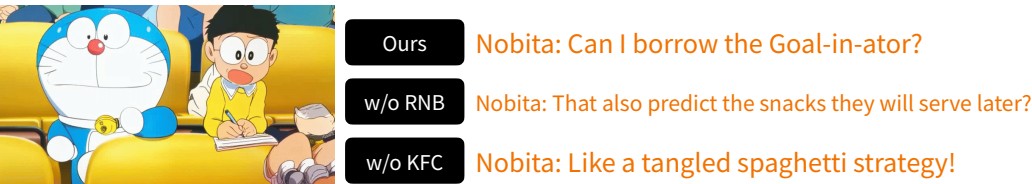

Figure 5: **Qualitative Ablation Results on Dialogue Coherence. (Full Version)** We showcase dialogue generated by our full model compared to ablated variants without the Recursive Narrative Bank (w/o RNB) and without Keyframe Conditioning (w/o KFC). The absence of RNB leads to disrupted narrative flow across scenes, while removing KFC yields contextually irrelevant or less grounded utterances. These results highlight the importance of both modules in producing coherent, character-consistent dialogue.

CLIPScore from 27.4822 to 25.3061, and DTW increases from 16.2412 to 22.4406. These declines confirm that grounding utterances in visual context is critical for generating coherent, character-consistent dialogue that aligns with the scene.

Next, we evaluate the impact of speaker conditioning in speech synthesis. Removing reference-based character embeddings leads to a significant rise in DTW—from 16.2412 to 49.6302—indicating a loss of prosodic consistency and character identity. Comparisons with other synthesis models such as Bark (16.4668) and Tacotron 2 (16.4484) show that while these methods offer reason-

able performance, our reference-guided approach produces more expressive and character-aligned speech.

We also conduct an end-to-end comparison by plugging our dialogue and speech components into different video generation frameworks. When combined with Mochi or Vlogger, overall quality degrades noticeably: BERTScore drops below 0.01, BLEU falls below 0.3, CLIPScore and DTW also worsen. These results suggest that despite strong language and audio generation, weaker video backbones limit overall storytelling quality. This demonstrates the importance of tight integration between motion modeling and narrative components.

To assess narrative coherence, we ablate the Recursive Narrative Bank (RNB), which maintains a memory of prior scenes and utterances. Without RNB, BERTScore declines to 0.0094 and BLEU to 0.5268, even though CLIPScore and DTW remain stable. This indicates that RNB primarily enhances linguistic continuity and long-range consistency in dialogue, which are essential for maintaining character development across scenes.

In summary, each component—visual grounding, speaker conditioning, and narrative memory—plays a distinct and complementary role in the system. The qualitative differences in Figure 5, along with quantitative improvements across all metrics, underscore the effectiveness of our integrated design.

## A.5 RECURSIVE NARRATIVE BANK AND SCRIPT THEORY

The Recursive Narrative Bank (RNB) is designed to enable coherent, character-driven dialogue generation in long-form multimodal storytelling by drawing on structured representations of memory inspired by cognitive science. Rather than functioning as a traditional memory buffer that passively recalls past utterances, RNB serves as a role-aware, temporally recursive scaffolding mechanism that simulates human-like conversational behavior over time. This distinction is essential: what makes dialogue compelling in extended narratives is not the surface-level repetition of past phrases, but the ability to produce behaviorally and emotionally consistent responses that evolve with the scene, role, and intention.

This principle is grounded in Script Theory Schank & Abelson (2013); Bower et al. (1979); Wilensky (1983), a foundational framework in cognitive psychology that models how humans interpret and produce behavior in structured situations. According to Script Theory, people rely on internalized "scripts"—sequences of stereotyped events and role expectations—to understand what to say, when to say it, and how to act in social contexts. For instance, a script for dining in a restaurant involves roles (customer, waiter), actions (ordering, serving), and expectations (e.g., the bill comes after dessert, not before). Dialogue within such scripts is not retrieved verbatim but generated based on the evolving context and the agent's role within it. The recursive structure of RNB operationalizes this by maintaining separate, character-specific dialogue histories that are re-injected into prompt templates in a way that mirrors how scripts are mentally activated and updated by humans during interaction.

Each RNB prompt is thus not a flat history window, but a structured invocation of a script fragment: it conditions the current scene on the immediate visual context (e.g., keyframe and action prompt) and the speaker's evolving narrative state. The inclusion of scene-level grounding ensures that the model's responses are not just temporally coherent but also visually relevant, maintaining alignment between what is said and what is shown. Moreover, by explicitly maintaining separate narrative banks for each character, the system supports differentiated behavioral trajectories—allowing one character's tone to escalate while another's remains calm, consistent with how individuals behave differently within the same scene.

This approach stands in contrast to generic prompt memory methods that treat history as unordered or speaker-agnostic. RNB is structured around the core dimensions of script-based modeling: temporality, role-awareness, and goal-consistent progression. Its recursive update mechanism ensures that the memory bank evolves over time without growing unbounded, allowing dynamic narrative control while remaining compatible with stateless large language model APIs. This makes RNB particularly well-suited to real-world generative systems, where persistent fine-tuned memory is unavailable but coherence remains critical.

By explicitly modeling narrative evolution through recursive, structured, character-aware prompts grounded in visual context, RNB enables zero-shot dialogue generation that reflects how humans produce situated language—not from scratch, but from structured expectations embedded in unfolding events. This theoretical grounding offers not only interpretability and generalizability but also a cognitively motivated lens for designing narrative-capable generative architectures.

To enhance clarity, we provide a formalized description of the end-to-end process used to generate character-consistent dialogue from scene prompts and visual context.

Let $(p_1^{(t)}, p_2^{(t)})$ denote the structured prompt pair at scene timestep $t$. These are concatenated to form the full prompt:

$$p^{(t)} = p_1^{(t)} + \text{`` . ''} + p_2^{(t)}. \tag{7}$$

Given the generated video clip $x_v^{(t)}$ for this prompt pair, we extract the middle frame:

$$I^{(t)} = \texttt{SampleFrame}(x_v^{(t)}, t = T/2), \tag{8}$$

where $T$ is the total number of frames.

We then compute a high-level visual semantic representation:

$$c^{(t)} = \texttt{SceneFeat}(I^{(t)}), \tag{9}$$

where `SceneFeat` is implemented using a pretrained BLIP captioning model. The result is a natural language caption aligned with the scene's visual content.

To ensure continuity across scenes, we define a Recursive Narrative Bank $\mathcal{H}_t$ as a temporally recursive memory:

$$\mathcal{H}_t = \{x_d^{(t-1)}, x_d^{(t-2)}, \ldots, x_d^{(t-N)}\}, \tag{10}$$

where each $x_d^{(i)}$ is a dialogue utterance generated at timestep $i$, and $N$ defines the memory window (e.g., $N = all$ in our implementation).

This memory, along with the current scene and visual embedding, is embedded into a structured input for the language model:

$$\texttt{Input}^{(t)} = [\texttt{Scene}]\, p^{(t)} \;\|\; [\texttt{Image}]\, c^{(t)} \;\|\; [\texttt{DialogueMemory}]\, \mathcal{H}_t. \tag{11}$$

The character-specific dialogue for the current scene is then generated via:

$$x_d^{(t)} = \texttt{LLM}(\texttt{Input}^{(t)}), \tag{12}$$

where `LLM` is a pretrained stateless large language model (e.g., GPT-4o). Only a single character speaks per turn, and speaker role is determined externally by prompt scheduling.

Finally, the narrative bank is updated recursively:

$$\mathcal{H}_{t+1} = \texttt{Truncate}(\mathcal{H}_t \cup \{x_d^{(t)}\}), \tag{13}$$

where `Truncate` enforces the memory limit $N$ by removing the oldest entry if necessary.

**Cognitive Perspective.** From the perspective of Script Theory Schank & Abelson (2013); Bower et al. (1979); Wilensky (1983), each structured prompt $\texttt{Input}^{(t)}$ simulates a localized fragment of a behavioral script. Rather than relying on rote memory, the model is guided by structured expectations derived from evolving visual and narrative cues. The separation into [Scene], [Image], and [DialogueMemory] reflects the key components of human-scripted interaction: situational setting, perceptual input, and role-consistent behavioral priors. This decomposition enables zero-shot stateless generation while preserving coherent narrative flow.

A.6 CONVERSATIONAL SPEECH GENERATION WITH RESIDUAL VECTOR QUANTIZATION

Conventional text-to-speech models directly map textual input to audio but often fail to reproduce the variability of conversational prosody. To overcome this limitation, we follow a residual vector quantization (RVQ) framework that represents continuous waveforms as discrete tokens, enabling

transformer-based modeling of both text and audio in a shared space AI (2024b). Two types of tokens are used: *semantic tokens*, which encode phonetic and linguistic content in a speaker-invariant manner but act as a prosodic bottleneck, and *acoustic tokens*, which preserve fine-grained attributes such as timbre, identity, and rhythm via RVQ. While semantic tokens provide a compact high-level abstraction, acoustic tokens are crucial for reconstructing high-fidelity and natural-sounding speech.

Let the text sequence be $T = \{t_1, \ldots, t_n\}$ and the conversational history be $A = \{a_1, \ldots, a_m\}$. The backbone transformer autoregressively models the zeroth-level codebook $k_0$ as

$$p(k_0 \mid T, A) = \prod_{t=1}^{n+m} p(k_{0,t} \mid k_{0,<t}, T_{\leq t}, A_{\leq t}), \tag{14}$$

where $k_0$ captures semantic and coarse prosodic structure. A lightweight decoder then reconstructs the higher-level residual codebooks $\{k_1, \ldots, k_{N-1}\}$ conditioned on $k_0$:

$$p(k_{1:N-1} \mid k_0) = \prod_{i=1}^{N-1} \prod_{t} p(k_{i,t} \mid k_{i,<t}, k_{<i}, k_0). \tag{15}$$

Here, each level $k_i$ refines acoustic resolution by conditioning on both its own history and all lower-level codebooks. Because RVQ imposes sequential dependence across levels, a *delay-pattern scheme* is employed in which higher codebooks are temporally offset to ensure conditioning on lower-level predictions. This improves fidelity but increases the time-to-first-audio, scaling linearly with the number of codebooks $N$.

To balance expressivity and efficiency, the Conversational Speech Model (CSM) separates modeling into two parts: a multimodal backbone for $k_0$ and a smaller decoder for $\{k_1, \ldots, k_{N-1}\}$. Generated acoustic tokens are autoregressively fed back into the backbone until an end-of-token symbol is reached, yielding coherent conversational speech. Text tokens are produced via a LLaMA tokenizer, and audio tokens are derived from a split-RVQ tokenizer that outputs one semantic and $N-1$ acoustic codebooks at 12.5 Hz.

Training such models presents a severe computational burden because the effective batch size is $B \times S \times N$, where $B$ is the batch size, $S$ the sequence length, and $N$ the number of RVQ levels. To mitigate this, we adopt a *compute amortization* strategy in which the backbone is trained on all frames for $k_0$, while the decoder is updated only on a random subset $\mathcal{F}' \subset \mathcal{F}$ of frames (with $|\mathcal{F}'|/|\mathcal{F}| = 1/16$). The loss function is thus

$$\mathcal{L} = \sum_{f \in \mathcal{F}} \mathcal{L}_{k_0}(f) + \sum_{f \in \mathcal{F}'} \sum_{i=1}^{N-1} \mathcal{L}_{k_i}(f), \tag{16}$$

where $\mathcal{L}_{k_i}(f)$ is the cross-entropy loss for predicting codebook $k_i$ at frame $f$. Empirically, this amortized scheme reduces memory and training cost without degrading perceptual quality.

Finally, to evaluate contextual speech generation, we employ both objective and subjective measures. Objective metrics include word error rate (WER) and speaker similarity (SIM), which saturate at near-human performance, as well as newly introduced benchmarks such as homograph disambiguation (e.g., distinguishing lead /lɛd/ vs. /liːd/) and pronunciation consistency across multi-turn speech. Subjective evaluation is conducted via Comparative Mean Opinion Score (CMOS) studies, where listeners compare model outputs against human recordings both with and without conversational context. While results show no significant difference without context, humans are consistently preferred when context is included, indicating that conversational prosody remains an open challenge. Limitations also remain in language coverage (primarily English) and the inability to fully capture higher-level turn-taking structures, though scaling model size and dataset diversity shows consistent improvement.

## A.7 COMPUTATION TIME AND MEMORY CONSUMPTION

Table 5 provides a breakdown of computation time and GPU memory consumption for our full system and its subcomponents. The full pipeline consumes approximately 31.21 GB of memory and takes 869 seconds to process a batch of stories, reflecting the combined cost of visual grounding, dialogue generation, and expressive speech synthesis.

To better understand individual module efficiency, we separately measure:

| Module | Step | Memory Consumption | Computation Time |
|---|---|---|---|
| Text2Story Kang et al. (2025) | Inference | 31.21 GB / 80.0 GB (29,767 MiB) | 869 sec |
| Natural Language Module (NLM) | Inference | 0.00 GB / 80.0 GB (0 MiB) | 6.46 sec / story (avg) |
| Speech Synthesis | Inference | 4.78 GB / 80.0 GB (4,567 MiB) | 4.04 sec / story (avg) |

Table 5: **Computation Time and Memory Consumption Analysis.** We report detailed computation statistics of the full model and its key submodules during inference, evaluated on an NVIDIA H100 GPU (80GB). The full pipeline integrates vision-language grounding, narrative modeling, and speech generation.

- **Natural Language Module (NLM)** – responsible for character-consistent dialogue generation based on prompt pairs and visual features. It consumes negligible GPU memory (0 MiB) and takes **6.4552 seconds per story on average**, totaling **83.92 seconds** for a representative case (Shrek & Donkey in San Francisco).

- **Speech Synthesis** – performed using a reference-guided speech generation model. It consumes **4.57 GB GPU memory** and requires **4.0389 seconds per story on average**, totaling **59.62 seconds** for the same case.

Although one may question whether lighter solutions like Mochi or Vlogger are preferable given their faster inference time (e.g., 126 sec total for Mochi), such comparisons overlook the quality-performance trade-off. Our full model is specifically optimized for coherent narrative flow, persona-consistent dialogue, and emotionally expressive speech. We justify the computational overhead through end-to-end evaluation, where our system significantly outperforms ablated or simplified baselines across automated and human preference metrics (see Table 1, Figure 4).

In summary, our pipeline demonstrates a strong balance between memory efficiency, inference time, and qualitative output. This makes it not only scalable but also effective for real-world storytelling applications where coherence, fidelity, and expressivity are essential.

## A.8   DATASETS (VIDEO GENERATION, AUDIO GENERATION)

We provide a structured benchmark for multimodal story generation across diverse narrative settings using familiar animated characters. This benchmark consists of multiple scene-level video clips, each represented by paired prompts: one describing the scene (*Prompt 1*) and another specifying the action (*Prompt 2*). Our benchmark is designed to test both the visual and auditory fidelity of generated narratives in character-driven storytelling.

**Video Generation.**   Our video generation dataset includes four themed narrative scenarios: **Urban Exploration in San Francisco**, **Nightlife in Las Vegas**, **Outdoor Cooking Show**, and **Sports Reporting Commentary**, all featuring familiar animated character pairs such as *Shrek & Donkey*, *Doraemon & Nobita*, and others. Each story comprises 11–13 sequential scene-action pairs (22–26 prompts in total), reflecting smooth scene transitions and character continuity. The prompts describe everyday actions (e.g., walking, sitting, looking) as well as location-specific interactions (e.g., cooking, cheering, jogging), making them well-suited for evaluating narrative coherence and multimodal alignment.

The **San Francisco** sequence begins with two characters arriving at the airport, retrieving their suitcase, and exploring iconic city landmarks including the Painted Ladies, Palace of Fine Arts, and the Golden Gate Bridge. The narrative concludes with a tranquil moment at Battery Spencer during sunset. The prompts highlight contextual changes in transportation (airplane, SUV, cable car), movement (walking, jogging), and visual engagement (gazing, admiring the view), facilitating fine-grained temporal generation.

In contrast, the **Las Vegas** narrative focuses on nighttime entertainment and visual spectacle. Starting with a walk along the Las Vegas Strip, two characters experience the Bellagio Fountain, interact with a slot machine, and later visit Fremont Street. This scenario emphasizes vivid lighting conditions, expressive reactions, and physical interactions, which are critical for evaluating temporal coherence and spatial attention in video generation.

The **Outdoor Cooking Show** features two characters at a forest campsite as they go through the process of preparing hot dogs. The narrative includes gathering ingredients, lighting a fire, and enjoying the meal. This instructional and grounded setup enables the assessment of stepwise procedural generation and causal alignment between actions and objects.

Finally, the **Sports Reporting** scenario depicts characters acting as soccer commentators in a stadium. Two characters provide play-by-play analysis, react to match events. This setting demands precise modeling of conversational rhythm, character roles, and referential language grounded in visual context, testing the system's ability to maintain long-term speaker consistency and context awareness.

**Prompt Format.** Each scene is defined by a prompt pair $(p_1, p_2)$, where $p_1$ establishes the setting and $p_2$ specifies the action. For example:

- `prompt_san_francisco_shrek_V_2122:`
  *["The sun begins to set over the Pacific Ocean", "Shrek and Donkey are standing"]*

- `prompt_vegas_shrek_V_1314:`
  *["Shrek and Donkey press a button on a slot machine in Las Vegas at night", "Shrek and Donkey are sitting"]*

Each full narrative includes 11–13 such prompt pairs (per setting), resulting in 22–26 scene-specific inputs per video. These are used to condition both video diffusion and dialogue generation models.

**Audio Generation.** To generate character-consistent speech, we use short voice clips publicly available on YouTube, released by official movie or studio channels. Specifically, our dataset includes two audio samples representative of expressive and emotionally charged dialogue by Shrek and Donkey, respectively. These clips serve as reference prompts for synthesizing conversational speech across scenes.

- `conversational_a (Shrek, 10sec):`
  *"We? Donkey, there's no we. There's no our. There's just me and my swamp. And the first thing I'm gonna do is build a 10-foot wall around my land."*

- `conversational_b (Donkey, 10sec):`
  *"Yes, I was talking to you. Can I just tell you that you was really great back there, man? Those guards thought they was all that. Then you showed up and bam!"*

- `conversational_a (Doraemon, 8sec):`
  *"Yep, first, the materials. Do you have any plastic lying around? Dump them in the mecha-maker."*

- `conversational_b (Nobita, 9sec):`
  *"We can make a ship with this thing? What do you mean? I got all these old toys I don't play with anymore."*

- `conversational_a (Tom, 12sec):`
  *"I'm Tom. You think I'm a dummy? Hey you little pipsqueak! How come you never spoke before!"*

- `conversational_b (Jerry, 8sec):`
  *"I'm Jerry. You said it, I didn't. There was nothing I wanted to say that I thought you'd understand."*

- `conversational_a (Minions Kevin, 5s):`
  *"Okay Doamato rapita ra polka moba ratriba findoreba bas."*

- `conversational_b (Minions Bob, 9sec):`
  *"Hm, uh, okay, okay, rakika, rebibas, Tony, prato, Tom, usaka, decrease, puratino."*

These reference clips are extracted from well-known scenes in the *Shrek* franchise and other iconic series and are used strictly for academic and non-commercial research. The usage complies with the United States **Fair Use Doctrine**, which permits limited use of copyrighted material for purposes such as research, teaching, and scholarship. We ensure that:

- The **reference clips** used as inputs are short excerpts (each under 30 seconds) taken from publicly available YouTube videos released by official sources. These are not redistributed or reused directly in our outputs, and they are not used in a manner that competes with the original work.

- These clips are used solely to **guide the prosody and expression** of our speech synthesis system. The **final generated audio** is newly synthesized and does not contain or replicate the original audio segments.

- The synthesized voices approximate character tone and emotion but avoid reproducing identifiable voiceprints or actor likenesses, thus minimizing ethical and legal concerns.

Our generated audio is an expressive approximation, not an imitation. It preserves the **persona and rhythm** of the character while avoiding the reproduction of identifiable voiceprints. This approach minimizes ethical and legal concerns while enabling consistent, role-aware voice synthesis across narrative scenes.

### A.9 ADDITIONAL ANALYSES: VOICE CONSISTENCY, PERSONA MODELING, AND ROBUSTNESS

**Voice Consistency Across Characters.** To further address reviewer concerns regarding occasional inconsistencies in generated speech, particularly for characters with monotonic delivery and low lexical diversity, we present a quantitative analysis of the reference voice prompts. Table 6 summarizes key lexical and acoustic properties across a diverse set of character types.

| Character | Unique Words | Voiced Phonemes | Pitch Std (Hz) | Pause Ratio | Duration (sec) |
|-----------|--------------|-----------------|----------------|-------------|----------------|
| Shrek | 23 | 70 | 57.77 | 1.00 | 10.00 |
| Donkey | 38 | 118 | 83.79 | 1.00 | 10.00 |
| Doraemon | 23 | 70 | 106.73 | 0.98 | 8.37 |
| Nobita | 24 | 59 | 146.51 | 0.94 | 9.81 |
| Kevin | 9 | 33 | 79.80 | 1.00 | 5.24 |
| Bob | 11 | 40 | 139.87 | 1.00 | 9.47 |
| Tom | 28 | 73 | 127.81 | 1.00 | 12.68 |
| Jerry | 18 | 57 | 136.64 | 1.00 | 8.16 |

Table 6: **Reference Prompt Statistics.** Voice prompts differ significantly in lexical richness, prosodic dynamics, and temporal structure. High pitch variability does not necessarily correlate with expressive or consistent speech style; utterances with low lexical variety and near-continuous voicing (pause ratio $\approx 1.0$) tend to sound more monotonic. This suggests that consistent speech perception depends on the interplay between lexical diversity, phoneme variation, and rhythmic structure—not pitch alone.

**Voice Consistency Across Characters: Metric Descriptions.** Each column in Table 6 is computed to quantify the linguistic and prosodic variability of the reference voice prompts:

- **Unique Words:** The number of distinct alphabetic tokens in the transcript, obtained using the NLTK tokenizer. This reflects lexical diversity, which contributes to the perceived richness and individuality of a character's speech.

- **Voiced Phonemes:** The number of voiced phonemes (e.g., /b/, /m/, /a/) in the transcript, extracted using the `phonemizer` library with the espeak backend (en-us). This metric captures phonetic variety and articulation complexity.

- **Pitch Std (Hz):** The standard deviation of the estimated fundamental frequency (F0), computed using WORLD vocoder methods (`pyworld.harvest` and `stonemask`). Higher values indicate greater prosodic variation and expressive tone.

- **Pause Ratio:** The proportion of silent frames in the audio, based on short-time energy computed with a sliding window (frame length = 2048, threshold = 0.01). A ratio near 1.0 indicates continuous voicing with few pauses, often perceived as monotonic delivery.

- **Duration (sec):** Total length of the prompt audio, calculated as the number of samples divided by the sampling rate.

These metrics jointly characterize the structure and expressivity of each prompt. We observe that characters with high pitch variance but low lexical diversity and minimal pauses (pause ratio $\approx 1.0$) tend to exhibit more frequent identity drift over long conversations. This suggests that speaker consistency is influenced not just by acoustic features, but also by linguistic variety and temporal structure. Future work may explore multi-turn prosody conditioning or speaker-aware memory to improve stability for such monotonic speech patterns.

**Character Persona Modeling.** Our system does not rely on static personality templates or trainable embeddings. Instead, character traits are dynamically inferred from the interplay of:

- the scene and action prompts,
- keyframe image captions (BLIP),
- and recursively accumulated dialogue history (via RNB).

This flexible mechanism allows for emergent behaviors and contextually appropriate responses, even for unseen or sparsely referenced characters like Tom and Jerry—who, as you may recall, rarely speak at all. In fact, their only widely known speaking moment comes from the delightfully controversial 1992 film Tom and Jerry: The Movie, which we bravely use as our sole reference. Rather than a limitation, we consider this an extreme zero-shot challenge: can the system generate character-faithful dialogue for two icons of silence? While the movie may have divided fans, our pipeline passed the test—producing speech that's chaotic, emotionally erratic, and somehow still on-brand. Just like Tom and Jerry themselves.

**Memory Length and Long-form Generation.** The Recursive Narrative Bank (RNB) retains *all* prior utterances by default (N = all), ensuring long-range dependency modeling and character continuity. Each dialogue turn remains compact (less than 100 characters), resulting in manageable prompt lengths. We have successfully generated stories with over 13 prior utterances without encountering token limits or degradation. This design supports scalable storytelling over extended sequences without truncation or loss of coherence.

**Robustness to Model Substitution.** Our modular pipeline decouples each stage (video, dialogue, speech) using intermediate natural-language representations. When we substitute the visual module (e.g., replacing Text2Story with Mochi or Vlogger), downstream components remain stable. As illustrated in Figure 5, quality degrades only locally (e.g., visual-semantic alignment) without cascading failures. This architecture supports component-level improvements and domain transfer without retraining the full pipeline.

