# OpenReview forum: "Character-Driven Narrative Generation for Scene-Based Video Synthesis"
_ICLR.cc/2026/Conference — ICLR 2026 Conference Withdrawn Submission_

### Official Review · Reviewer_BSh7 · 2025-10-31

**Soundness:** 3
**Presentation:** 2
**Contribution:** 2
**Rating:** 4
**Confidence:** 4

**Summary:**

The paper proposes a training-free, modular pipeline that turns scene-based prompts into character-driven dialogue and expressive speech for story videos. A story model makes short scene clips; a vision-language encoder (BLIP) captions a keyframe to ground context; a large language model generates persona-consistent lines using a Recursive Narrative Bank (RNB) that tracks per-speaker dialogue history; then a reference-driven TTS renders character-conditioned voices. The authors report quantitative gains over ablations (e.g., removing visual grounding or RNB) and strong preferences in a small human study, claiming more natural, context-aware, and consistent dialogue across multi-scene narratives.

**Strengths:**

- The proposed task is interesting. Character-driven dialogue is important but ignored in current research community.
- Works by composing existing video, VLM, LLM, and TTS blocks; easy to swap components and domains
- Diverse story settings (urban exploration, cooking, sports) suggest robustness to theme/style changes.
- Users favored the full pipeline in most comparisons, highlighting perceived naturalness and persona consistency.

**Weaknesses:**

- Human study is small (15 participants), scenarios center on cartoon characters, and absolute automatic scores (e.g., BLEU/BERTScore) are low and hard to interpret without stronger baselines or references to external datasets.
- Using one representative keyframe can miss actions/emotions that happen elsewhere in the clip, limiting dialogue-scene alignment for dynamic moments.
- The work reads more like a practical how-to for combining off-the-shelf VLM/LLM/TTS tools than a research contribution: there are no new training objectives, architectures, or inference algorithms. The “Recursive Narrative Bank” is essentially a prompt/history heuristic, not a principled modeling or decoding advance. As a result, the paper’s technical novelty is thin, focusing on component wiring rather than new methods.

**Questions:**

My major concern is the third point of these weaknesses.

---

### Official Review · Reviewer_STLo · 2025-10-31

**Soundness:** 2
**Presentation:** 3
**Contribution:** 2
**Rating:** 4
**Confidence:** 4

**Summary:**

This paper addresses a gap in storytelling video generation: the lack of coherent, character-driven dialogue and speech. The authors propose Action2Dialogue, a modular, training-free pipeline that synthesizes character-specific dialogue and expressive speech directly from scene-level text prompts.
The system takes a sequence of prompt pairs as input, where each pair defines a scene's setting and a character's action. The pipeline then performs three main functions:
Scene Visualization: An existing story generation model (e.g., Text2Story) creates the visual video clip.
Dialogue Generation: A large language model (LLM) generates dialogue. This is the core contribution. The LLM is conditioned on the input prompts, visual features from a representative keyframe (extracted via a model like BLIP), and a novel dialogue history mechanism.
Speech Synthesis: A reference-driven voice synthesis model renders the generated text into expressive speech using a character's voice sample.
The paper's primary component is the Recursive Narrative Bank (RNB), a speaker-aware, temporally-structured memory. This mechanism provides the LLM with the dialogue history, enabling it to generate utterances that are consistent with a character's persona and evolving narrative context across multiple scenes. The authors demonstrate through automated metrics and a human subject study that their full pipeline, particularly the visual grounding and the RNB, produces significantly more natural, coherent, and character-consistent narratives than ablated baselines.

**Strengths:**

1. The paper tackles a clear and currently underserved problem in generative AI. While video generation has advanced, most outputs are "silent films." This work provides a concrete framework for adding a crucial dimension of emotional depth and narrative realism through character-driven dialogue.
2. The proposed method is training-free and does not introduce new model architecture, but propose modular composition of existing SOTA components (video generation, VLM, LLM, TTS).
3. The paper is well-written and easy to follow. The problem statement is clear, the proposed method is described logically, and Figure provides an excellent overview of the entire pipeline. The authors clearly delineate their contributions.

**Weaknesses:**

1. Missing Lip-Sync / Audiovisual Synchronization: The most significant weakness is the lack of lip synchronization. The paper generates a video clip and a separate audio track. While it calls this a "multimodal video narrative," the result is effectively a video with a character-specific voice-over, not a video of a character speaking. This omission feels like an incomplete solution to the problem of "character-driven dialogue" in a visual medium. A truly integrated system would need to either generate the video with the correct mouth movements or apply a post-processing lip-sync model. This limitation should be more prominently discussed. Video and dialogue are generated by two separate, unaligned systems. The video model generates visuals from (p1, p2), while the LLM generates text from (p1, p2, c, Ht). There is no mechanism to enforce that the generated dialogue precisely matches the specific generated visuals.
2. Weak Visual Grounding: The system's entire visual understanding of a scene hinges on a single, representative middle frame. This is a very lossy representation of a dynamic video clip. If a key action described in the prompt occurs at the beginning or end of the clip, the middle frame might be uninformative (e.g., just showing Donkey on the ground), leading to poorly-grounded dialogue. A more robust approach might use a video-language model to encode the entire clip's temporal dynamics.
3. The paper's related work section is missing a critical line of research: speech-driven video generation (or talking head/portrait animation). This field directly addresses the paper's (unsolved) final step of creating a speaking character.
Highly relevant works like

EMO: Emote Portrait Alive -- Generating Expressive Portrait Videos with Audio2Video Diffusion Model under Weak Conditions

Hallo3: Highly Dynamic and Realistic Portrait Image Animation with Video Diffusion Transformer

omnihuman-1: rethinking the scaling-up of one-stage conditioned human animation models

mocha: towards movie-grade talking character synthesis

**Questions:**

On Lip-Sync: The lack of lip-sync is the most apparent gap. See weaknesses

---

### Official Review · Reviewer_2pG7 · 2025-11-01

**Soundness:** 2
**Presentation:** 2
**Contribution:** 2
**Rating:** 4
**Confidence:** 4

**Summary:**

The paper argues that recent video generation models exhibit significant limitations in generating character-driven dialogue and speech, which diminishes their narrative realism. To address this limitation, the authors propose a training-free modular pipeline. This pipeline comprises three stages: video-based prompt augmentation, dialogue generation via a Recursive Narrative Bank (RNB), and reference-based speech generation. The proposed pipeline demonstrates significant improvements at each corresponding stage.

**Strengths:**

- As shown in Table 1, each module outperforms its corresponding baseline on the designated task
- Quantitative comparisons reveal that the generated dialogues are more narratively consistent and realistic than those produced by the baseline models.

**Weaknesses:**

- Presentation has some drawbacks:
   * The formulation of $p$ at the beginning of Section 3.2 is unclear. Specifically, the meaning of the $". "$ symbol within the equation is confusing.
   * Figure 4 combines three subfigures, which renders each one too small to be clearly inspected.
   * All quantitative experiments are consolidated into Table 1. However, this table conflates results from different tasks, which hinders clarity. Separating these results by task would be much clearer. For example, metrics such as BERTScore, BLEU, and CLIPScore are included for the 'Speech Generation & Speaker Conditioning' task, where they appear to be irrelevant.
- The experimental design is also debatable.
   * The end-to-end comparison is limited, as it included only Vlogger and Mochi. However, other capable models exist that can generate video and speech concurrently. It is recommended that the authors expand their comparison to include these models, evaluating the quality of the generated voice.
- The paper seems to primarily contribute a speech generation method tailored for the video generation context, rather than on a holistic, end-to-end video generation pipeline. The title may be somewhat misleading, as it may implie a broader scope.
   * As a following up, if the contribution is a speech generation module seemlessly integrated with video generation models, it is recommended that the authors apply it to diversefoundation video generation models. Conducting such comparisons would be necessary to verify the method's generalizability and effectiveness across different architectures.

**Questions:**

- The specific data structure of the BNR remains somewhat unclear. The paper describes it as a _role-conditioned, temporally structured memory for each character_ (L.102), but this description is quite abstract. Could the authors provide examples of a BNR instance? Does this 'role-conditioned' and 'structured' format imply a JSON-like object where keys correspond to character roles, like the structured prompt formats used by systems like the OpenAI API?

---

### Official Review · Reviewer_9H5r · 2025-11-02

**Soundness:** 2
**Presentation:** 3
**Contribution:** 2
**Rating:** 2
**Confidence:** 3

**Summary:**

The paper proposes a modular framework for character-driven narrative video generation that integrates scene prompts, large language models, and reference-based speech synthesis. Its central idea is the Recursive Narrative Bank (RNB), which maintains speaker-aware dialogue memory to enhance cross-scene coherence.

**Strengths:**

The work is well-structured, addresses an interesting and timely multimodal storytelling problem, and demonstrates clear engineering value through the integration of visual, linguistic, and auditory modalities.

**Weaknesses:**

1.The core innovation lies almost solely in RNB, while other components rely on existing powerful models.
2.The evaluation metrics are questionable: BERTScore≈0.067 and BLEU≈1.87 are extremely low for dialogue generation, suggesting the metrics are unsuitable for short, persona-driven utterances—thus the reported “+71.9% / +45.0% improvement” is largely a relative amplification.
3.Metric–task mismatch: comparing dialogue text with scene prompts or captions measures lexical overlap (“similar to the prompt”) rather than conversational quality, persona consistency, or emotional appropriateness.
4.Table inconsistencies: identical or near-identical metric values across methods raise doubts about statistical reliability and whether results were computed or reused correctly.
5.Using a single representative frame as the sole visual condition is risky—dynamic cues, temporal causality, and emotional transitions are lost; no multi-frame or video-encoder comparison is provided.
6.The RNB definition is inconsistent—the paper alternately describes a fixed window and “N = all”; it’s unclear how memory truncation or scaling is actually handled.
7.The speech-side evaluation is overly simplistic—using only pitch-contour DTW to assess expressiveness and consistency ignores crucial prosodic and perceptual factors such as speech rate, pause ratio, energy dynamics, formant structure, and rhythm–emotion coupling. Moreover, the paper lacks subjective listening tests and speaker-similarity analysis, which are standard for validating perceptual and identity-related quality.
8.The human evaluation is too small-scale and shallow—only 15 participants with 45 votes per setting. Although confidence intervals are reported, there is no mention of significance testing, inter-annotator agreement, or task-specific sub-scores (e.g., content relevance, tone, pacing, lip-sync, or persona consistency). These omissions make it difficult to rule out random preference or bias effects.

**Questions:**

see weaknesses

---

### Note · Authors · 2025-11-12

I have read and agree with the venue's withdrawal policy on behalf of myself and my co-authors.